# A Holistic Approach to Cooling System Selection and Injection Molding Process Optimization Based on Non-Dominated Sorting

**DOI:** 10.3390/polym14224842

**Published:** 2022-11-10

**Authors:** Janez Gotlih, Miran Brezocnik, Snehashis Pal, Igor Drstvensek, Timi Karner, Tomaz Brajlih

**Affiliations:** Faculty of Mechanical Engineering, University of Maribor, Smetanova ulica 17, 2000 Maribor, Slovenia

**Keywords:** conformal cooling, injection molding, tooling, additive manufacturing, numerical simulation, non-dominated sorting

## Abstract

This study applied a holistic approach to the problem of controlling the temperature of critical areas of tools using conformal cooling. The entire injection molding process is evaluated at the tool design stage using four criteria, one from each stage of the process cycle, to produce a tool with effective cooling that enables short cycle times and ensures good product quality. Tool manufacturing time and cost, as well as tool life, are considered in the optimization by introducing a novel tool-efficiency index. The multi-objective optimization is based on numerical simulations. The simulation results show that conformal cooling effectively cools the critical area of the tool and provides the shortest cycle times and the lowest warpage, but this comes with a trade-off in the tool-efficiency index. By using the tool-efficiency index with non-dominated sorting, the number of relevant simulation cases could be reduced to six, which greatly simplifies the decision regarding the choice of cooling system and process parameters. Based on the study, a tool with conformal cooling channels was made, and a coolant inlet temperature of 20 °C and a flow rate of 5 L/min for conformal and 7.5–9.5 L/min for conventional cooling channels were selected for production. The simulation results were validated by experimental measurements.

## 1. Introduction

One of the main problems in manufacturing thermoplastic injection molding tools for high volume production is temperature control of critical areas. The products and tools are becoming increasingly demanding in terms of design, functionality, esthetics, and process (granules, cycle times). Therefore, the production of cooling channels that direct the cooling medium to a specific location in the tool is very demanding [1,2,3]. To achieve the optimal tool temperature, the tool must be both cooled and heated, which is called temperature control of the tool. The task of the temperature control system is to maintain the temperature in the tool that is most suitable for the polymer to be injected and to keep the temperature as uniform as possible over the entire surface of the tool [4,5]. The quality of a polymer product requires maintaining a cycle with a constant, uniform temperature field in the tool and as uniform a progression of isotherms as possible to ensure uniform temperature control of each part of the product [6]. For simple products, it is easy to determine the course of the isotherms. For sophisticated products, however, the course of the isotherms is often too complex to determine without the help of numerical simulations.

Conventionally, flat engravings are temperature-controlled with longitudinally and transversely drilled channels, depending on the recommended spacing and size of the channels, which form a circuit for the cooling medium. It is also possible to mill a spiral pattern for the circuit, which is covered and sealed with an additional plate [7,8]. On the other hand, large flat shapes can become a major challenge when a flat surface and uniform wall thickness are required. In such cases, where there is a large surface area and many channels, the whole system must be divided into several individual circuits to ensure adequate temperature control [9].

For taller cores and box molds where there is sufficient space for holes along the core, baffles or bubblers are used for temperature control [10]. Baffles or bubblers are also used when drilling cross holes in the engraving plate is not possible. Baffles or bubblers are made by drilling, with the diameter of the holes ranging from Ø 6 to Ø 23 mm. A sheet metal or screw insert is placed in the hole to direct the flow of the cooling medium. Correct installation of the insert is important because if it is incorrectly placed, the baffle will lose its function. Baffles or bubblers are connected to each other by a cross hole to form a circuit.

Thin cores pose a particular problem because they do not provide enough space for cooling circuits and are difficult to cool due to their slenderness [11]. To cool thin cores, holes must be drilled into the core, which compromises their strength, functionality, and lifetime. There are two standard options for cooling thin cores: cooling with fountains with tube overflow or with special inserts. In both cases, a very precise hole must be drilled into the core due to the limited space and tolerances required. A fountain is inserted into the hole, and the cooling medium then flows around the core. If using an insert, a copper tube is inserted into the hole, which is in contact with the cooling medium to dissipate heat. Fountains with outer diameters ranging from Ø 2 to Ø 8 mm and inserts with outer diameters ranging from Ø 2 to Ø 10 mm are used [12,13].

These are the standard methods for temperature control of simple shapes. They can be combined with each other, and they do not pose any problem for manufacturing with conventional technologies. They ensure efficient cooling and optimal service life of the tool. 

For more inaccessible and varied parts of tools that cannot support a drilled cooling circuit, materials much more thermally conductive than ordinary tool steel can be used. Highly thermally conductive tool steels with conductivities of up to 70 W/mK are often used when the tool has moving parts or is heavily loaded but does not have sufficient space for cooling circuits, as is the case in narrow, angled ejectors and tool inserts [14]. Another option is to use an alloy of copper and beryllium that achieves a thermal conductivity of 156 W/mK, which is about three times that of ordinary tool steel [15]. The CuBe alloy can be used if the components of the tool are stiff enough, installation is not a problem, and heat dissipation is the most important thing to consider. For efficient cooling, the CuBe alloy parts must be in contact with the temperature-control medium. When using CuBe alloy cores, we encounter problems such as the difficulty of installation, low strength, high price, and even toxicity [16].

If the above methods fail, conformal cooling, which has recently become popular, could be a solution. Conformal cooling means that the temperature control channels follow the shape of the part and, ideally, always run at the same distance from the surface of the part [17,18]. The thermodynamics of temperature control are complex and have a decisive influence on the course, size, and shape of the channels when designing the tool. Using numerical simulations, we can determine the appropriate way to control the temperature of a product or tool by modeling the shape of the product, the tool, and the temperature control system and then setting the process parameters based on data of the processed polymer, the cooling medium, the material of the tool, and processing conditions. In many cases, this means that we cannot determine the optimal temperature control arrangement until after the tool has been designed. Some recent advances have been reported regarding sophisticated methodology or even automatic cooling circuit generation based on topological optimization [19,20,21,22,23,24].

In creating optimal channels, we encounter many technological obstacles. Designing an injection molding tool thus involves a search for a compromise between the ideal shape of the channels, their technological manufacturability, and the feasibility of the produced tool. Conformal cooling, which involves the use of additive manufacturing technology, is a solution to this problem, as it allows manufacturing of complicated cooling channel layouts and cross-section designs to enhance heat transfer. Comparison between conventional and conformal cooling has demonstrated significant benefits to conformal cooling in terms of reducing cycle time, increasing product quality, and achieving an even temperature distribution over the entire part surface [25,26,27,28,29,30]. However, economic concerns need to be considered when selecting an appropriate tool design. Conformal cooling leads to an increase in the tool cost due to intricate cooling channels, which may not be justified for small production batches. However, it also reduces per piece production cost by improving production rate and reducing the cooling time. Thus, a breakeven analysis must be performed [31].

Different additive manufacturing techniques are appropriate for tool manufacturing. Low-cost techniques, such as fused filament fabrication (FFF) and selective laser sintering (SLS), for fabricating injection molds were studied, and considerable reductions in tool service life and cooling efficiency were reported, making low-cost tool design inappropriate for mass production [32,33]. Tools fabricated using a powder-bed fusion (PBF) process are more suited for high-production-volume tools. Selective laser melting (SLM) technology is one of the most-used methods of layered construction of metal products [34,35,36]. Beyond fabrication of simple conformal channels, the additive manufacturing process enables the introduction of special shapes of cooling channels and even fabrication of cell structure with high accuracy, which could achieve a significant reduction in cooling time [37,38]. However, the additive manufacturing process also has some important disadvantages that affect the performance of the injection molding process. Smooth outer surfaces cannot be achieved, and if the inside of the channel is rough, both cooling performance and tool life are negatively affected. The results of mechanical tests showed that SLM manufactured H13 parts have worse mechanical properties compared with conventional materials, but these improved after residual stresses were relieved by hardening [39].

Metallic materials that can be used for additive manufacturing of tools include tool steels; stainless steels; CoCr alloys; Ni, Al, and Ti alloys; and bronze alloys. The development of a novel hybrid powder-wrought alloy steel for tool inserts has also been reported [40]. The service life of 316L stainless steel SLM inserts was found to be suitable for high-volume production [41], although the reported batch size was still lower than that which can be expected from conventionally produced tool steel molds, and the production process takes longer. Additionally, mechanically mixed nylon-12/copper composite powders were investigated and compared as alternatives to produce injection molding tools with conformal cooling channels [42].

The performance of the injection molding process could also be improved using optimization techniques. Numerical simulations combined with statistical approaches or non-deterministic algorithms are often used to optimize the parameters of the injection molding process to ensure the desired product quality and reduce the cycle time. Tool design optimization mainly involves the design of the cooling system and selection of the tool material with the aim of achieving effective cooling, reducing the manufacturing cost of the tool, and ensuring the expected lifetime of the tool [43,44,45,46]. Notably, however, a holistic optimization approach that considers tool design, tool manufacturing, process parameters, and product quality to support the decision of whether to manufacture a tool with conventional or conformal cooling has not yet been presented.

The objective of this research is to apply state-of-the-art technologies to complement conventional tooling technologies with additive manufacturing to reduce cycle time, achieve the required product quality, reduce tool manufacturing time and cost, and achieve material properties that provide the required tool service life for high-volume production. A novel tool-efficiency index for holistic process optimization is introduced to evaluate tool manufacturing time, cost, and expected service life. In an example, the entire injection molding process is evaluated at the tool design stage using four criteria, one from each stage of the process cycle. These criteria are tool design, tool production, process control, and product quality, and the goal was to produce a tool with effective cooling that enables short cycle times and ensures good product quality. It was shown that by using numerical simulations and additive manufacturing, it is possible to control the temperature of critical areas in the tool and prevent overheating of the tool, which resulted in reduced process downtime while maintaining good properties and long tool life.

This article is organized as follows: in Section 2, the design of the conventional and conformal cooling system is presented, and a numerical simulation is performed. A holistic decision support algorithm with non-dominated sorting is presented, and the objective criteria are explained. In Section 3, the numerical simulation results are presented, the non-dominated solutions are obtained, and (based on the non-dominated solutions) a Pareto frontier is generated. The decision to make a hybrid tool is discussed, and the manufacturing process is explained. The tool is tested in the injection molding process, and the process and product quality are evaluated. The simulation results are verified using a thermal imaging camera. Section 4 concludes the article.

## 2. Materials and Methods

The goal of the study was to optimize cooling, ensure stable injection molding conditions, achieve consistent part quality, shorten cycle time, ensure required tool lifetime, ensure low manufacturing costs, and provide the ability to produce the core insert quickly. In addition to some of the conflicting objectives, constraints are imposed on the product, the tool, and the process, meaning the problem can be considered an optimization problem with multiple objectives. To achieve the desired results, the optimization objectives must be considered during tool design, tool manufacturing, and injection molding, which is a long process performed by multiple departments and subcontractors. Therefore, a holistic strategy is required already in the tool design phase, which is usually defined in the virtual environment with the support of CAD/CAM/CAE software. 

For this study, geometric CAD models and CAE simulations were created using Catia, Visi, and Moldflow software. Catia was used to create CAD models of the product and the tool components, Visi was used to create CAM models for manufacturing of the tool, and Moldflow was used to simulate the injection molding process. The tool core insert was manufactured using hybrid technology: partly with conventional machining and partly with additive manufacturing. A Battenfeld 6500 injection molding machine was used for production, and a FLIR E6 thermal imaging camera was used to verify thermal conditions in the tool during production. The entire process with the critical decision on the selection of the cooling system is schematically shown in Figure 1.

### 2.1. Product Requirements and Tool Design Constraints

Some technological requirements for the injection molding process were established in advance. The product shape, the plastic material, the injection location, the number of cavities, and the productivity of the tool were predetermined and had to be considered as constraints in the design of the tool. Because the required dimensions of the product were within narrow tolerances, the tool was very demanding to manufacture. The product batches were also exceptionally large, which meant that the tool would be in constant use. Therefore, the process had to be stable, and the cycle had to be as short as possible. 

Due to the shape of the product, which was a fog light housing with an irregular shape (Figure 1), the tool was functionally very demanding. The product was characterized mainly by the concave chamber for electrical connectors in the upper part of the housing. The chamber represented the critical area of the product in terms of production by injection molding, and it increased the complexity of the tool by requiring an insert with moving parts. This made it difficult to efficiently control the temperature of the critical area of the product, and it made the overall production process less reliable. Apart from the critical area, the product had three mounting plates and features for adjusting the height of the fog lights on the outer walls, which further increased the complexity of the tool. Another feature that required special attention was the uneven wall thickness of the product, which varied up to a maximum thickness of 6 mm. The bounding box of the product was 106 × 90 × 82 mm^3^.

The tool was predefined to have four cavities. Each of the cavities had four external openings that covered almost the entire product and three internal openings in the core insert (Figure 1). This meant there were many moving parts that had to be temperature controlled to achieve a stable injection molding process. Most of the temperature control had to be done on the ejector side, the part of the tool with the external openings, where the main problem was that there is not much space for the cooling channels. However, there were no restrictions preventing the use of conventional cooling techniques. For temperature control inside the product, the three internal openings in the core posed the largest problem. Without the internal openings, the core could easily be temperature controlled using baffles, but with the openings it was not possible to direct the cooling medium to the critical area using conventional temperature control techniques.

### 2.2. Cooling System Design

The cooling system can be divided into two subsystems: the ejection-side cooling system and the core-side cooling system. Mainly for economic reasons, a conventional cooling system is preferred to a conformal cooling system for industrial tools. Therefore, the entire cooling system was originally designed as a conventional cooling system and later upgraded with conformal channels to improve the cooling of the critical area. In this study, both cases were considered in a comparative study.

#### 2.2.1. Ejection-Side Cooling System Design

The temperature on the ejection side of the tool was controlled by conventional cooling techniques (Figure 2). Blind channels are unavoidable when the cooling system is made using conventional manufacturing techniques, but they were not modeled as part of the cooling system because they do not contribute to cooling. On the ejector side, a conventional cooling system was designed to prioritize the critical areas and provide more cooling to the sensitive areas of the tool.

In Figure 2, the green-colored and azure-colored channels were designed to cool the left side, right side, and critical area of the product. Both channels had a diameter of Ø 8 mm and used baffles with a diameter of Ø 5 mm to reach the openings in the critical area of the product to increase heat extraction. Both channels and baffles could be drilled to their final shape. The emerald-colored and orange-colored (Figure 2b) channels were designed to cool the top, back, and critical area of the product. Both channels were Ø 10 mm in diameter and could be drilled to their final shape. In addition to the critical area, the thickest parts of the product also need more cooling. Variations in the wall thickness of the product can be problematic, especially for those parts of the product that have greater wall thickness. In the case of the fog light housing, there were some large variations in wall thickness, with a significant increase in wall thickness at the front. The most thickened areas were accessible with conventional cooling channels and could be cooled with the cyan-colored channel (Figure 2c). The channel had a diameter of Ø 8 mm and could be drilled to its final shape.

#### 2.2.2. Core-Side Cooling System Design

The core with three internal openings, which may not be adequately cooled with conventional cooling methods, was the main subject of this study (Figure 3). To find the optimal design of the cooling system, two cooling system layouts were prepared. The first study case (a) was used to evaluate a core design with cooling channels that could be fabricated using conventional methods. This case was referred to as the conventional cooling case and guaranteed fast fabrication and low manufacturing cost of the core. It also guaranteed good mechanical and heat transfer properties, but due to the complexity of the product, effective temperature control may not be achieved. The second study case (b) was used to evaluate a core design in which the upper part of the core with integrated conformal cooling channels was fabricated through additive manufacturing. This case was referred to as the conformal cooling case.

For the conventional cooling case, presented in Figure 3a, the 6 mm diameter cooling channels were designed to be drilled to their final shape. Due to the functional requirements of the core, the channels could only be placed in the lower part of the product and do not reach the critical area of the product. This design can be a major limitation for the targeted short cycle time. 

For the case of conformal cooling, presented in Figure 3b, the inlet and outlet channels with a diameter of Ø 6 mm were placed in the lower part of the core and designed to be drilled to their final shape. The conformal cooling channel, with a diameter of Ø 5 mm, was connected to the inlet and outlet channels and passed laterally along the inner opening of the product through the critical area of the core. The conformal cooling channel was designed to be manufactured by additive manufacturing. The diameter of the channel was optimized with the aim of, first, increasing the Reynolds number and thus the efficiency of heat dissipation of the conformal cooling channel and, second, reducing the surface roughness as effectively as possible by using an abrasive medium, which increases the efficiency of heat dissipation and extends the life of the tool [47]. The layout of the conformal cooling channels was designed to follow the shape of the product and to be evenly spaced from the walls of the product. The design allowed for easy maintenance and the possibility of surface finishing to further increase tool life.

### 2.3. Numerical Simulation of the Injection Molding Process

To learn how efficient the critical areas of the product could be cooled and how this affected the cycle time, the quality of the product, the tool manufacturing time, cost, and service life, three different analysis types were performed: a cooling analysis to evaluate the cooling performance, a fill analysis to evaluate the cycle time, and a warp analysis to evaluate the product quality. To compare the two cooling approaches (conventional and conformal), two case studies were created. The first study was set up to simulate an injection molding process in which the cooling channels were produced by conventional methods without cooling the critical area (Figure 3a). In this approach, the tool was manufactured by conventional methods and consisted of the same material. Therefore, a numerically less extensive BEM (boundary element method) analysis was used. The second study was set up to simulate an injection molding process in which the cooling channels were produced by additive manufacturing and the critical area of the product was additionally cooled (Figure 3b). In this approach, the tool components that were in contact with the product consisted of two different materials. To accurately simulate the injection molding process for the second study, a FEM (finite element method) analysis with a detailed tool model was used.

Computational meshes of the product, tool, and of the conventional cooling channels were created for both simulation cases (Figure 4). 

In Figure 4a, the product mesh model with the injection location is shown. The mesh models for the product were completely the same for both study cases. In (Figure 4b), the tool mesh model is presented, which consisted of 21 components and the additive manufactured top part of the tool core insert presented in (Figure 4c). The detailed tool model was only used in the case of conformal cooling. In the case of conventional cooling, a simplified sheet-element-based tool model was used. The conventional cooling channels were completely modeled as beam elements, as shown in (Figure 4d). This was done to increase computational efficiency, as it was observed that three-dimensional channel models did not influence the model accuracy but significantly increased computational resources. The conformal cooling channel is the only channel that was modeled with a three-dimensional mesh (Figure 4e) to capture the temperature conditions more exactly in the critical area of the product and to account for the difference in material properties of the core base and core insert top part.

The mesh model details are presented in Table 1.

#### 2.3.1. Material Properties

The tool was made of tool steel 1.2343, which is a common tool material for injection molding. It is a high-alloy hot-work tool steel with high thermal shock resistance, good heat resistance, high thermal conductivity, and toughness. 

The upper part of the core insert was made of maraging steel 1.2709. Because the upper part of the core was produced by additive manufacturing, the thermal and mechanical properties depended not only on the temperature but also on the additive manufacturing production parameters [48,49]. 

The thermoplastic material selected for the fog light housing was Lupoy HR5007AB from the manufacturer LG Chemical. The material belongs to the group of ABS-PC materials, which are widely used in the automotive industry due to their good processing, mechanical and thermal properties, and cost efficiency compared with PC. 

Water was used as the cooling medium. The thermal properties of the used materials given in Table 2 are to be understood as estimated values for the critical temperatures.

When selecting tool materials for the hybrid tool, it is important to choose materials with compatible mechanical properties. Mechanical properties affect the quality of the product, the machinability, the manufacturing process, and, finally, the life of the tool and the frequency of maintenance. Maraging steel 1.2709 was selected for the core upgrade because it has similar mechanical properties to tool steel 1.2343. Because of its low carbon content, maraging steel 1.2709 is very weldable and resistant to crack formation when the material is rapidly heated and cooled during the SLM process. After SLM production, the product achieves a hardness of 33–37 HRc, a manufacturing accuracy of ±50 μm, a surface roughness of Ra 5 μm and Rz 28 μm, a relative density of almost 100 %, a density of 8–8.1 g/cm^3^, and a tensile strength of 1200 MPa.

A major disadvantage of the layer-by-layer manufacturing process is the anisotropy of the material. This is reflected in lower mechanical properties in the build-up direction of the product. Hardening can significantly improve the mechanical properties of the product. After hardening, a tensile strength of 2100 MPa and a hardness of 50–56 HRc are achieved, while the product remains dimensionally stable, and the negative effects of anisotropy are reduced. Both after the SLM process and after hardening, the product can be further processed by machining, electro erosion, welding, polishing and surface treatment. 

#### 2.3.2. Physical Models and Processing Parameters

For conventional cooling, a BEM analysis was performed. For conformal cooling, a FEM analysis was performed. Both analyses used the same physical and processing parameters (Table 3). 

The automatic filling time and the boundary condition values were used for an initial calculation to determine the cycle time. For the simulations of the optimization study, the calculated filling time was then used as a variation basis in combination with variable boundary conditions. 

### 2.4. Holistic Decision Support for the Choice of a Cooling Strategy

The decision regarding the cooling strategy affects the entire product life cycle and must be made at the tool design stage. There are a variety of channel design options, so a systematic design approach is required. Once the final tool design is determined, a detailed optimization of the process parameters can be performed to find the optimal molding conditions for the entire product life cycle. 

In the following, a novel holistic approach to cooling strategy selection is presented that evaluates the impact of different cooling strategies on various criteria that may be directly related to product quality, the molding process, tool design, and tool manufacturing. To avoid considering too many objectives, only one objective was selected at each stage. The selected objective, which was considered as a product quality criterion, was warpage since high product deformation is usually the main cause of a defective product. The process-oriented objective was cycle time, as shortening the cycle time is highly desirable from an economic and flexibility point of view. The tool design-oriented objective was average product surface temperature, which is a good indicator of cooling efficiency of the product. The manufacturing-oriented objectives were tool manufacturing time, manufacturing cost, and tool life, which were combined into a new tool-efficiency index. 

#### 2.4.1. Evaluation of the Tool-Efficiency Index

For tool design and manufacturing-oriented optimization, a review of the literature and market research was conducted, and a tool performance evaluation system for each criterion was introduced. In our case, the relative parameter performance was evaluated. The values can also be quantified for any specific case.

The first criterion considered was tool life. When additive manufacturing methods are used to manufacture tools, lower production volumes are reported [16,31,41]. Therefore, the tool core insert manufactured by SLM was evaluated as worse in terms of tool life than the conventionally machined insert. In terms of manufacturing costs, the core insert produced by SLM was also evaluated as worse because it requires an additional step in the manufacturing process. For the tool studied, the additional price for four core inserts ranged from EUR 900 to EUR 1800 depending on the subcontractors. In terms of manufacturing time, the SLM-manufactured insert was also rated as worse because the extra step in the manufacturing process takes more time. Moreover, a potentially more serious problem is that additive manufacturing machines are not profitable for many toolmakers, meaning they must work with subcontractors. In our case, working with the subcontractor extended the manufacturing time by more than two weeks, which meant a loss of time, and, in the case of a damaged insert, it means a long production shutdown to make a replacement.

In order to quantify the tool-efficiency index and evaluate it using the non-dominated sorting method, the parameters $n_{cyc}$, $mf_{cost}$, and $mf_{time}$ were assigned the values –1 or 1, according to their ranking in Table 4, where “–“ equals −1 and “+” equals 1.

The tool-efficiency index $\mu_{tool}$ can now be introduced as follows:(1)$$\mu_{tool}=\left(n_{cyc}+mf_{cost}+mf_{time}\right)$$

The evaluation of the tool-efficiency index results in $\mu_{tool}=-3$ for conventional core and $\mu_{tool}=3$ for conformal core.

#### 2.4.2. Evaluation of Average Product Surface Temperature, Cycle Time, and Warpage

To investigate the effects of process variables on average product surface temperature ($T_{s,avg}$), cycle time ($t_{cyc}$), and warpage ($\Delta_{w,a}$), a series of numerical simulations were performed. The simulations were performed as a parametric study in the case of conventional cooling and as a face-centered cubic DOE study in the case of conformal cooling. Because the channel for conformal cooling was the only channel modeled as a three-dimensional element, additional boundary conditions were required, resulting in two additional variables. Therefore, to reduce the number of calculation cases, the conformal cooling study was organized as a DOE study instead of a parametric study. Process variables are presented in Table 5.

For the conventional and conformal cooling cases, an initial simulation was performed with the same parameter settings to determine the optimal cycle time, which served as the basis for the parametric and DOE studies. The final parametric study resulted in 27 calculation cases, and the DOE study resulted in 43 calculation cases.

#### 2.4.3. Decision Support by Non-Dominated Sorting

Considering all objectives, the objective vector can now be expressed as follows:(2)$$x=\left(\begin{array}{c} \begin{array}{c} T_{s,avg}\\ t_{cyc} \end{array}\\ \begin{array}{c} \Delta_{w,a}\\ \mu_{tool,neg} \end{array} \end{array}\right)$$
where $T_{s,avg}$ is the average surface temperature of the product, $t_{cyc}$ is the cycle time, $\Delta_{w,a}$ is the warpage, and $\mu_{tool,neg}$ is the negative value of the tool-efficiency index. The negative value of the tool-efficiency index was used in the sorting process because the sorting algorithm looks for minimum values of the results to order them from best to worst, but in the case of the tool-efficiency index, higher values are considered better.

With respect to the objective vector x, it is now possible to find non-dominated solutions and form a Pareto frontier from which a holistically optimal solution can be selected before making the final decision on tool design, manufacturing, and subsequent injection molding parameters.

Generally, a multi-objective problem can be formulated as:(3)$$\min\left[f_{1}\left(x\right),\;f_{2}\left(x\right),\ldots ,f_{n}\left(x\right)\right];\;n>1;\;x\in S$$
where f is a scalar objective function and S is the feasible set of decision vectors, generally defined as:(4)$$S=\left\{x\in R^{m}:h\left(x\right)=0,g\left(x\right)\geq 0\right\}$$

The space in which the objective vector belongs is called the objective space, and the image of the feasible set under F is called the attained set:(5)$$C=\left\{y\in R^{n}:y=f\left(x\right),x\in S\right\}$$

A vector $x^{*}\in S$ is said to be Pareto optimal for a multi-objective problem if all other vectors x∈S have a higher value for at least one of the objective functions $f_{i}$, with i=1, …, n, or have the same value for all the objective functions.

### 2.5. Experimental Validation of the Injection Molding Process

A Battenfeld 6500 injection molding machine was used for production. The optimized parameters from the simulation were used to set up the machine. Once the correct processing conditions were stabilized, the thermal conditions in the tool were tested using a FLIR E6 thermal imaging camera with a measurement range of −20 °C to 550 °C. Measurements by the thermal imaging camera proved useful for monitoring tool and core temperatures after ejection, to locate possible overheated zones, and to obtain information about temperature levels and the difference between individual cavities [50,51]. Each cavity and the critical areas of the cavities were checked. The cycle time was also monitored. The measurements were performed in an industrial environment during the injection molding process.

## 3. Results and Discussion

### 3.1. Numerical Simulation Results

The calculated time to reach the ejection temperature (Figure 5), the cycle time without opening the tool and the time for product ejection, was 25.41 s (Figure 5a) for the conventionally cooled tool and 21.13 s (Figure 5b) for the conformally cooled tool. The critical area was not the biggest problem for product cooling because the thick areas where the product features are combined cooled more slowly. Although conformal cooling is not designed to increase heat extraction from the thick areas, the thick areas still cooled down faster when conformal cooling was used, resulting in faster attainment of the ejection temperature and shorter cycle time.

In our case, heat extraction from the tool was more problematic than heat extraction from the product. The temperature of the tool indicated local overheating, which affected the cycle time and the deformation of the product. The recommended tool surface temperature should be within 10 °C for amorphous polymers such as ABS-PC [52]. As a general guideline, the smaller the temperature differences and the more uniform the tool temperature, the less deformation and stress on the product and the shorter the cycle time. The comparison between tool temperature results for conventionally and conformally cooled tools is shown in Figure 6. For the conventionally cooled tool (Figure 6a), the tool temperature is not in the 10 °C range, but it is for the conformally cooled tool (Figure 6b). For the conventionally cooled tool, the critical area was the hottest part of the tool and exceeded the maximum allowable temperature. This is undesirable because the tool can overheat, which increases the cycle time. In a conformally cooled tool, the temperature in the critical zone was much lower (about 40 °C), meaning the desired cooling effect was achieved by the conformal cooling channel. The only overheated zone was the area where the mounting plate is attached to the cylindrical part of the product, but the temperature was still below the maximum allowable tool temperature, and due to the reinforcing features, this area is not problematic for deformation.

The temperature of the product, just like the temperature of the tool, indicates critical areas, poor cooling efficiency, and the final quality of the product (Figure 7). The recommended temperature difference between the outside and inside of the product is 5 °C [52]. The greater the difference, the more deformed the product will be. Conventional and conformal cooling do not meet the recommendation, although for conformal cooling, only the area where the mounting plate is attached to the product is critical. By optimizing the boundary conditions for cooling, the cooling effect should be improved.

### 3.2. Optimization Study Results and Pareto Frontier

Non-dominated sorting was used to select from all simulation results, and a Pareto frontier was created (Figure 8). The blue dots in the plots represent the first Pareto frontier, while the red dots represent the results for all the remaining computational cases for reference. Because the problem is defined as a four-objective problem, it is presented in a three-dimensional diagram where the tool-efficiency index is excluded (Figure 8a) and in three two-dimensional diagrams where the individual objectives are plotted against the tool-efficiency index (Figure 8b–d). The best results were obtained for the conformally cooled tool, which had a tool-efficiency index of −3. The average product surface temperature, time to reach ejection temperature, and deflection results were all lowest in combination with the conformally cooled tool. However, since there are several non-dominated results on the Pareto frontier, none of them is ultimately the best, and a final decision has yet to be made.

Table 6 shows the Pareto frontier cases with results and variable values. Four cases are for conformal cooling and two cases are for conventional cooling. The lowest average surface temperature and shortest cycle time were found in case number 23 for conformal cooling. The deformation and cycle times were very similar in all cases both for conformal and conventional cooling. The largest difference was found for the average product surface temperature. 

In all cases, the time for injection, packing and cooling ($t_{i,p,c}$), and inlet boundary temperature for conventional cooling channels ($T_{c,B}$) have the most significant influence. This is because the maximum value of the time for injection and packing and cooling as well as the minimum value of the inlet boundary temperature for conventional cooling channels (20 °C) are reached in all cases. The second most important variable was the Reynolds number (*Re*), which was the same in five of the six cases and had its maximum allowable value. The boundary conditions for the conformal cooling channel did not have a significant influence.

To be able to select the best outcome and better understand the process, the simulation results for the cases on the Pareto frontier were compared. As mentioned in the previous section, the most important result was the product temperature, where a critical region was found at the junction between the mounting plate and the product.

The product temperature results for the non-dominated cases are shown in Figure 9. The most uniform product temperature was found in cases (Figure 9d) and (Figure 9f), which were the conformal cooling cases where the coolant inlet temperature was the highest while the flow rate did not seem to have a significant effect. Cases (Figure 9a) and (Figure 9b) were conventional cooling cases where the critical area was still not sufficiently cooled, which may lead to tool overheating. In cases (c) and (e), the critical area of the product was cooled too much, which caused the inside of the product to be much cooler than the outside. In all cases, the area where the mounting plate is connected to the cylindrical part of the product was now cooled much more effectively, which ensured a much more uniform product temperature.

The results of product deformation are shown in Figure 10. There were no significant differences between the cases considered, and the maximum deformation was less than 1 mm, which is within the desired tolerance limit.

### 3.3. Manufacturing of Hybrid Tool Cores

To avoid problems with local overheating of the tool and due to the predicted short cycle times and acceptable product deformation, the conformal cooling system was chosen for production. The tool core insert was manufactured using hybrid technology. It was produced partly with conventional machining and partly with additive manufacturing [53,54,55].

The tool core insert was fabricated in two steps. Two separate CAD models were created to support the fabrication process (Figure 11). In the first step, the model representing only the base of the core insert (without the upper part of the core insert) was modeled and used to prepare the CAM program for milling and drilling (Figure 11b). In the second step, the model representing the upper part of the core insert with an integrated conformal cooling channel was modeled and used to prepare the CAM program for additive manufacturing by the SLM process (Figure 11b). Due to the limited workspace of the SLM machine, two cores were modeled together on one support block, which was also used to define a reference coordinate system (Figure 11c).

The core insert base was made by milling from tool steel 1.2343. In the first step of the production process, a rough tool shape was machined with an addition of 0.5 mm of material on the walls for hardening. After hardening, the material reached a hardness of 50–52 HRc. In the second step of the production process, the reference surfaces were machined by milling and grinding to prepare the base for the upgrade with the SLM process. To establish the coordinate frame of the workpiece in the SLM machine, a reference surface and a hole were fine machined on the support block (Figure 12a). 

The core insert top was manufactured by additive manufacturing from material 1.2709. The EOSINT M280 machine with a 400 W ytterbium laser was used. SLM process parameters such as laser power, scanning speed, layer thickness, construction strategy, angle of inclination of the laser beam, and remelting influence the product quality as well as the speed and cost of production [56,57,58]. The original parameters of the machine, labelled MS1 Performance 1.0, were used. The thickness of each layer was 40 μm, the laser scanning strategy of the layer was linear and 45° with respect to the Y axis. On each subsequent layer, the scan direction was rotated 45° clockwise until the end of the program. The process of making one pair of inserts took about 7 h on the device. In addition, 4 h were used to prepare the CAD model, 2 h to prepare the CAM program, and 2 h to set up the machine. Two cores were produced in one machine job, which means that two machine jobs were required to produce all four cores. After SLM production (Figure 12b), the products had a hardness of approximately 35 HRC. To improve the mechanical properties, the inserts were hardened for 6 h at a temperature of 530 °C to achieve a hardness of 52–54 HRc.

### 3.4. The Injection Molding Process and Temperature Measurement Results

During the injection molding process, the actual temperatures of the critical areas were measured and compared with the simulation results (Figure 13). The temperature of each tool cavity was between 50 °C and 55 °C, which is in accordance with the recommendations and was consistent with the results of the simulation (Figure 13c). During the tests, the cooling of one cavity failed in one case. The thermal imaging camera image shows the temperature distribution along the ejection half of the tool. The second cavity in the upper right corner glows more than the other three due to overheating (Figure 13a). To continue the test, we had to wait until the tool cooled down sufficiently. After a few cycles, the tool overheated again, and the test failed. It was found that one of the channels was contaminated with milling chips. After the channels were cleaned, the core no longer overheated, and the test passed. The corresponding photo of the tool can be seen in (Figure 13b).

Another problem we observed was sink marks occurring at the thickest part of the product (Figure 14a). During the simulation, we noticed a much higher internal temperature in the region (Figure 14b), but the sink mark prediction was not able to predict the sink marks at this point (Figure 14c). After testing, it was determined that the sink marks did not occur regularly, and the slightly adjusted process parameters of simulation case 19 were adopted.

The finished fog light housing is shown in Figure 15a. The left and right fog light housings with the transparent insert piece are shown in (Figure 15b).

## 4. Conclusions

The objective of this research was to investigate the feasibility of conformal cooling and to optimize the injection molding process holistically by considering tool design, tool manufacturing, the injection molding process, and product quality as objective criteria. The state of research in conformal cooling showed that various processing parameters had already been optimized, but there was no study that considered the entire tool design, tool manufacturing, and injection molding process from a holistic optimization perspective. Because wrong decisions in the concept phase of tool development can have costly consequences, such as lower product quality, longer cycle times, high production costs of the tool, or high maintenance costs, it is important to consider the entire process already in the concept phase.

This study was conducted on a complex automotive fog light housing product that had a critical area where efficient cooling could not be achieved by conventional methods. This study also contained some predefined tool design constraints. A holistic approach was proposed to decide whether to use conventional or conformal cooling and to learn how this decision affects the entire process. The presented holistic approach involves considering a target objective at each stage of the injection molding process. In this way, a four-dimensional decision vector was created. In our case, the selected objectives were cycle time, warpage, product surface temperature, and a novel tool-efficiency index that reflects tool cost, tool manufacturing time, and expected tool service life. To investigate the effect of conformal cooling, two cooling system layouts were prepared: conventional and conformal. Cycle time, product deformation, and product surface temperature were evaluated by a series of numerical simulations prepared according to the parametric study and DOE methodology. In these simulations, the cooling system layout, the total time for injection, packing and cooling, and the coolant boundary conditions were defined as variables. The obtained results were ranked by non-dominated sorting, and a Pareto frontier was created to allow easy decision making for which simulation cases should be studied in more detail.

The results of the non-dominated sorting show that conformal cooling is superior to conventional cooling in all objectives except for the tool-efficiency index, where the conventional cooling system performs better in all criteria considered. This is due to the cheaper and faster production process, as the tooling components are produced using conventional manufacturing techniques, while the tooling components with integrated conformal cooling channels are produced using expensive and time consuming SLM technology. In addition, large scale production with SLM-manufactured tools is not yet common. In our case, a hybrid tool core insert was produced, which extended the production time by more than two weeks and increased the cost of the four core inserts by EUR 900. However, simulation results showed that product and tool temperatures were much more uniform in the cases with conformal cooling, which ensured stable injection molding conditions and dimensional stability of the product with short cycle times. The use of conformal cooling channels reduced the temperature in the critical area by 20 °C. In this sense, the benefits of conformal cooling outweighed the higher cost and time required to produce the conformally cooled tool, and the conformally cooled tool was selected for production. The process for making the tool core insert is explained in detail, and the selection of reference points is highlighted as a problematic process in the production of hybrid tools. Experimental testing of the tool confirmed the predicted temperatures, although during testing, one of the four cores overheated. Contamination with milling chips was found in one of the channels. Because of their simple shape, the conformal cooling channels were easy to clean, and the overheating problem was solved without causing additional downtime during the tool test procedure. During the test runs, sink mark damage that had not been predicted in the simulation was discovered on one of the products. Because this was a one-time event, slightly adjusted simulation-based process parameters were adopted for series production. 

The conformally cooled tool has many advantages over the conventionally cooled tool, but since SLM equipment is very expensive, most tool manufacturers cannot afford it. This forces outsourcing, which costs time and money. Our conclusion is that these challenges should not be avoided, as they represent progress and allow toolmakers to take a step forward in an environment where product requirements are increasing every day.

## Figures and Tables

**Figure 1 polymers-14-04842-f001:**
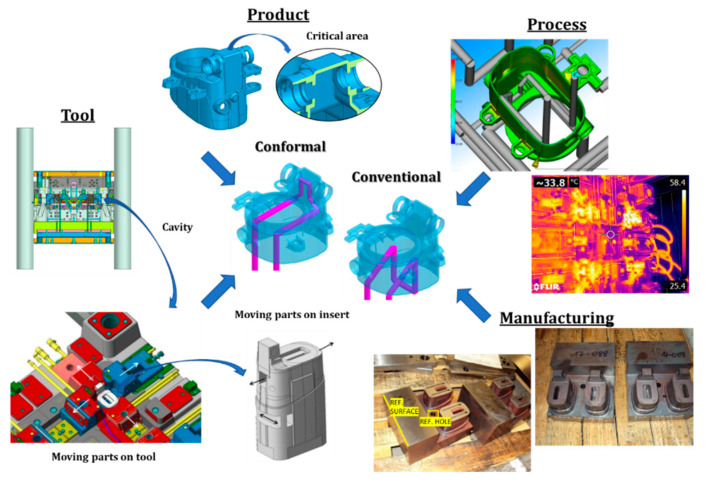
Challenges in selecting the cooling concept for injection molding.

**Figure 2 polymers-14-04842-f002:**
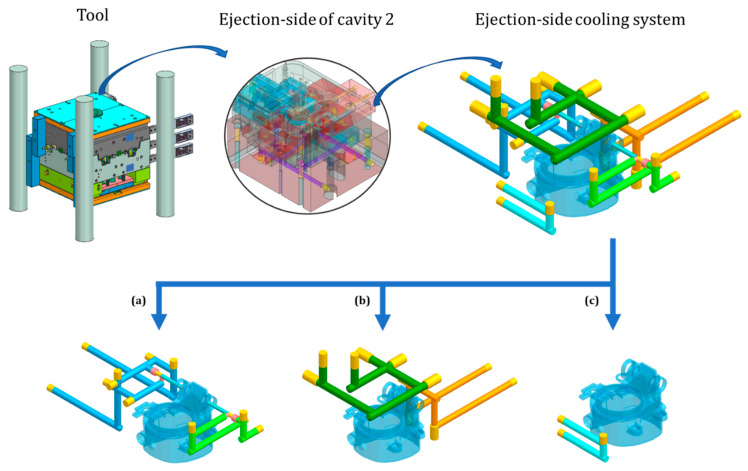
View of the tool showing details of the ejection side of cavity 2, the conventional cooling system, and individual conventional cooling channels with products (**a**–**c**).

**Figure 3 polymers-14-04842-f003:**
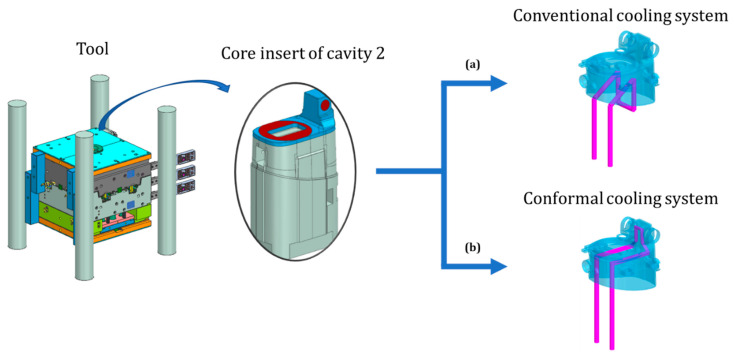
View of the tool showing a detail of the core insert of cavity 2 with a solution for conventional (**a**) and conformal cooling (**b**).

**Figure 4 polymers-14-04842-f004:**
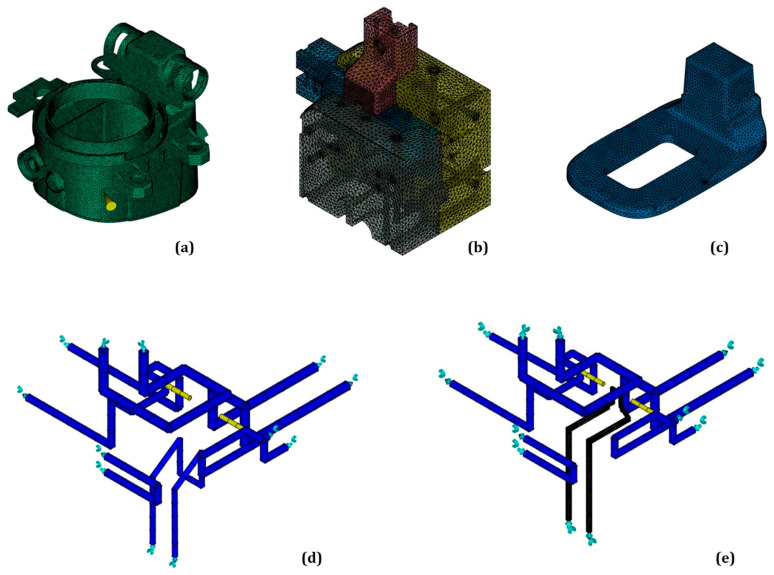
Mesh models of the product with injection location (**a**), the detailed tool (**b**), the top part of the tool core insert (**c**), the conventional cooling system with inlet and outlet boundaries (**d**), and the conformal cooling system (**e**).

**Figure 5 polymers-14-04842-f005:**
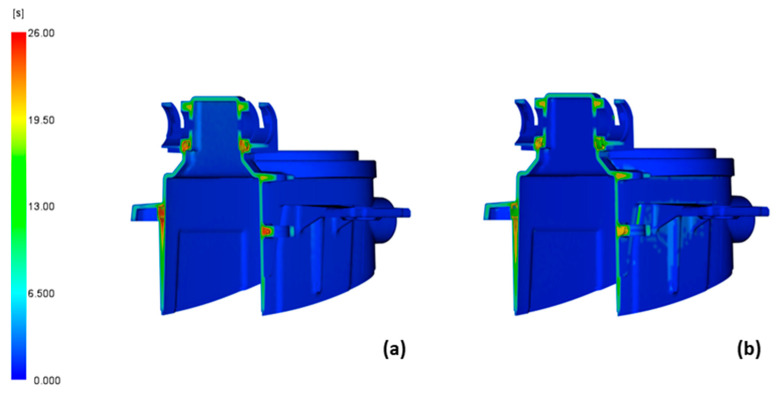
Time to reach the ejection temperature in the cross section at the critical area of the product calculated with conventional cooling (**a**) and with conformal cooling (**b**).

**Figure 6 polymers-14-04842-f006:**
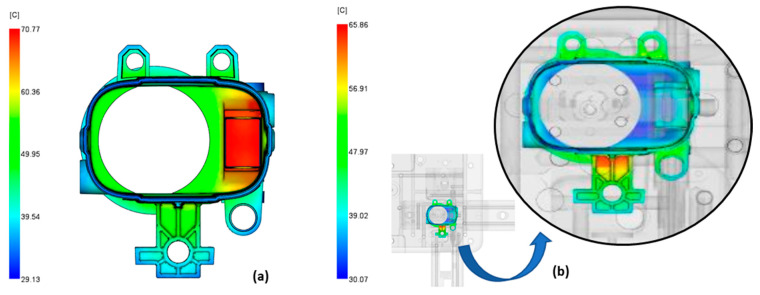
Tool–product interface temperature for conventionally cooled (**a**) and conformally cooled tool (**b**).

**Figure 7 polymers-14-04842-f007:**
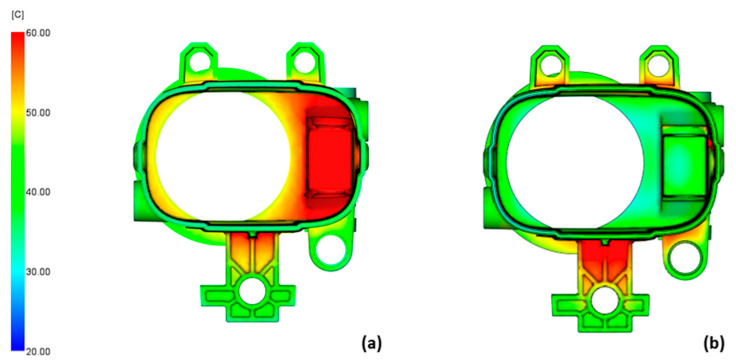
Temperature of the product for conventional cooling (**a**) and conformal cooling (**b**).

**Figure 8 polymers-14-04842-f008:**
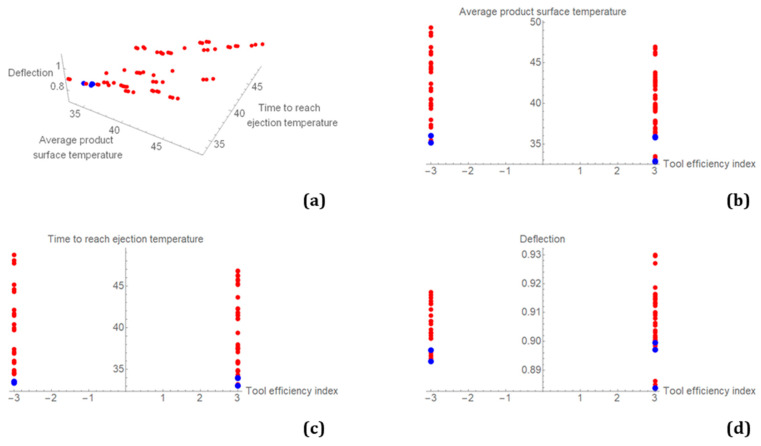
Three-dimensional Pareto frontier without the tool-efficiency index objective (**a**) and two-dimensional Pareto frontiers for individual objectives plotted against the tool-efficiency index (**b**–**d**).

**Figure 9 polymers-14-04842-f009:**
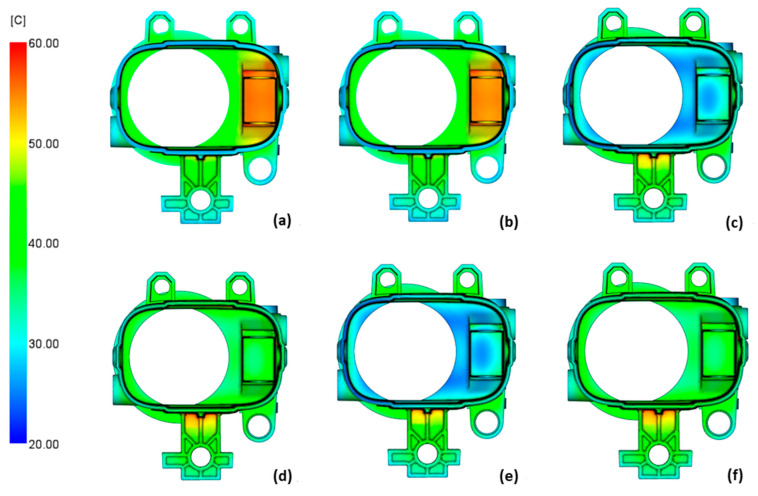
Product temperature results for conventionally cooled case 3 (**a**) and case 21 (**b**) and for the conformally cooled case 19 (**c**), case 20 (**d**), case 23 (**e**), and case 24 (**f**).

**Figure 10 polymers-14-04842-f010:**
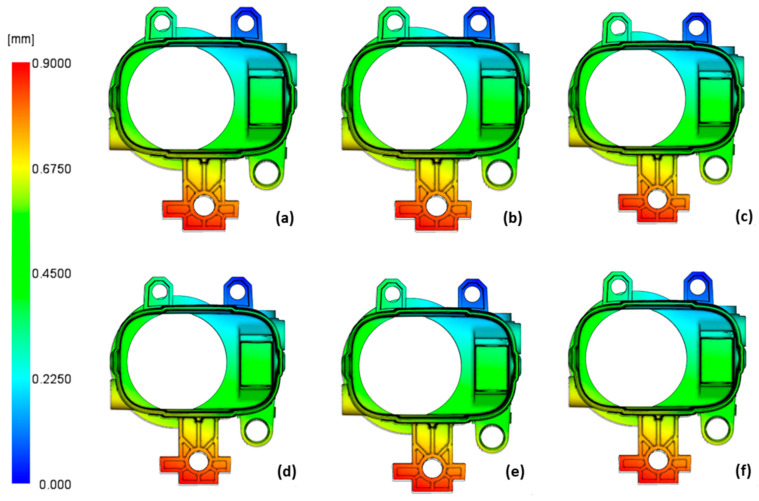
Product deformation results for conventionally cooled case 3 (**a**) and case 21 (**b**) and for the conformally cooled case 19 (**c**), case 20 (**d**), case 23 (**e**), and case 24 (**f**).

**Figure 11 polymers-14-04842-f011:**
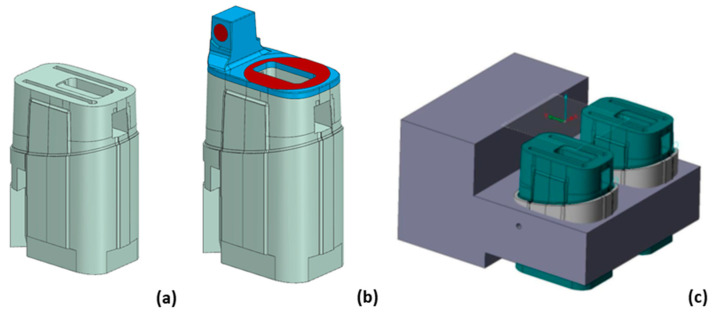
Core insert base model (**a**), model of the complete assembly of the core insert (**b**), and workpiece with reference coordinate frame for the SLM manufacturing process (**c**).

**Figure 12 polymers-14-04842-f012:**
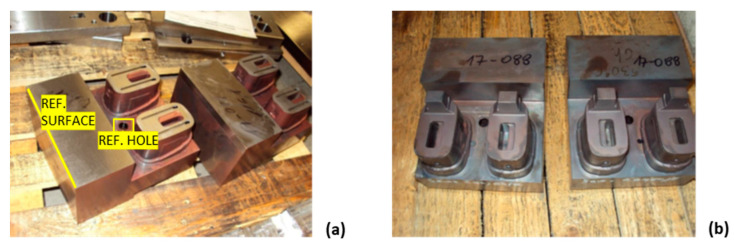
Tool core base after hardening with reference features for SLM (**a**), all four tool cores prepared for hardening after completed SLM upgrade (**b**).

**Figure 13 polymers-14-04842-f013:**
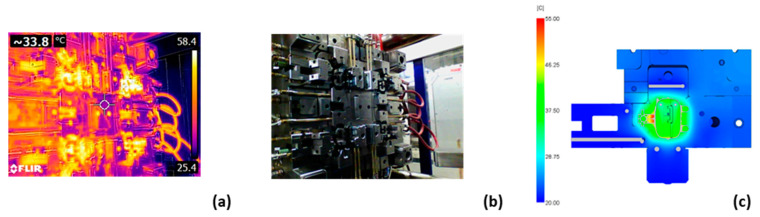
The thermal image of the tool reveals local overheating (**a**), the corresponding photo of the tool (**b**), and the tool temperature simulation results (**c**).

**Figure 14 polymers-14-04842-f014:**
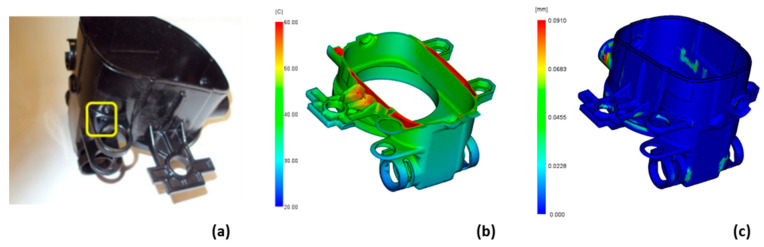
Sink mark due to material concentration on product (**a**) product temperature in the sink mark region (**b**) and sink mark prediction result (**c**).

**Figure 15 polymers-14-04842-f015:**
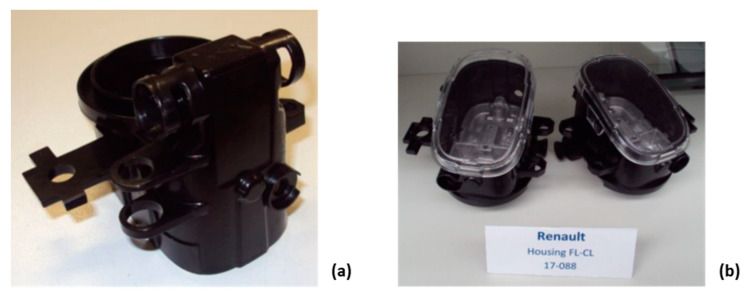
Finished product (**a**) and left and right product in fog light assembly (**b**).

**Table 1 polymers-14-04842-t001:** Comparison of mesh models for conventional and conformal cooling.

	Conventional Cooling	Conformal Cooling
Number of injection location nodes	1	1
Number of nodes	186,047	185,928
Number of product tetrahedral elements	1,020,561	1,020,561
Number of tool sheet elements	474	-
Number of tool tetrahedral elements	-	2,535,903
Number of tetrahedral tool core insert elements	-	151,944
Number of channel tetrahedral elements	-	392,577
Number of channel beam elements	114	87

**Table 2 polymers-14-04842-t002:** Thermal properties of the materials used.

Description	Symbol	Unit	Lupoy HR5007AB ^1^	Water ^2^	Tool Steel 1.2343 ^3^	Maraging Steel 1.2709 ^3^
Density	ρ	$\mathrm{kg}/{\mathrm{cm}}^{3}$	1003.60 – 1136.70	988	7800	8100
Specific heat capacity	$c_{p}$	J/(kg·K)	1352 – 2205	4180	460 – 550	400 – 470
Thermal conductivity	λ	W/(m·K)	0.20 – 0.24	0.64	25.30 – 27.20	14.20 – 19.00
Thermal expansion coefficient	α	$10^{-6}/\;K$	/	/	10.80 – 11.80	10.72 – 11.50

^1^ ρ melt state-solid state; $C_{p}$ at 51–265 °C; λ at 31–277 °C; ^2^ ρ, $C_{p}$, λ at 25 °C; ^3^ $C_{p}$ at 50–300 °C; λ at 20–350 °C; α at 100–300 °C.

**Table 3 polymers-14-04842-t003:** Simulation model control parameters.

Solver control	Solver	Coupled 3D
	pvT model	2-domain modified Tait
	Viscosity model	Cross-WLF
	Residual stress model	without CRIMS
Temperature control	Averaged within cycle	ON
	Ambient temperature	25 °C
	Tool temperature	70 °C
	Melt temperature	265 °C
	Ejection temperature	96 °C
Tool-melt heat transfer coefficient	Filling	5000 W/m^2^ °C
	Packing	2500 W/m^2^ °C
	Detached, cavity side	1250 W/m^2^ °C
	Detached, core side	1250 W/m^2^ °C
Filling control	Filling control type	Automatic
	Velocity/pressure switch-over	Automatic
Pack/holding pressure control	Profile	Constant
	Duration	10 s
	% of filling pressure	80
Warpage control	Small deflection	ON
	Use mesh aggregation	ON
Boundary conditions	Reynolds number	10,000
	Flow rate (3D channels)	10 L/min
	Inlet temperature (beams)	25 °C
	Inlet temperature (3D channels)	25 °C

**Table 4 polymers-14-04842-t004:** Conventional and conformal tool performance comparison.

	Conventional Insert	Conformal Insert
$n_{cyc}$	+	−
$mf_{cost}$	+	−
$mf_{time}$	+	−

$n_{cyc}$, tool lifetime; $mf_{cost}$, manufacturing cost; $mf_{time}$, manufacturing time.

**Table 5 polymers-14-04842-t005:** Process variables and variable limits for parametric study of conventional cooling and DOE study of conformal cooling.

	Conventional Cooling (Parametric)	Conformal Cooling (DOE)
Re	±50%	±50%
Q	/	±50%
$T_{c,B}$	±25%	±25%
$T_{c,D}$	/	±25%
$t_{i,p,c}$	±25%	±25%

Re, coolant inlet Reynolds number; Q, coolant flow rate, only a variable in case of conformal cooling; $T_{c,B}$, coolant inlet temperature; $T_{c,D}$, coolant inlet temperature, only a variable in case of conformal cooling; $t_{i,p,c}$, time for injection, packing, and cooling.

**Table 6 polymers-14-04842-t006:** The Pareto frontier calculation cases with variable values and results.

**Case Number**	Re **[−]**	Q **[L/min]**	$T_{c,B}$ **[°C]**	$T_{c,D}$ **[°C]**	$t_{i,p,c}$ **[s]**	$T_{s,avg}$ **[°C]**	$t_{cyc}$ **[s]**	$\Delta_{w,a}$ **[mm]**	$\mu_{tool}$ **[−]**
3	10,000	-	20	-	35	36.00	33.58	0.8930	−3
21	20,000	-	20	-	35	35.16	33.41	0.8970	−3
19	20,000	5	20	20	31	32.91	33.06	0.8971	3
20	20,000	5	20	30	31	35.90	33.98	0.8816	3
23	20,000	15	20	20	31	32.78	33.05	0.8995	3
24	20,000	15	20	30	31	35.80	33.95	0.8838	3

## Data Availability

The data supporting the results of this study are available from the corresponding author upon request.

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
