# Peer review of "A Holistic Approach to Cooling System Selection and Injection Molding Process Optimization Based on Non-Dominated Sorting"

_polymers, 2022, doi:10.3390/polym14224842_

Round 1
Reviewer 1 Report
Dear Authors,
First, I want to congratulate the authors on their research which is very well written and presented.
The research is focused on manufacturing of a fog light housing with an irregular shape by the injection molding process. A holistic optimization of the process is presented in terms of tool cooling operation by comparing the conventional and conformal one. Numerical simulation results of the time to reach the ejection temperature, temperature distribution, and product deformation are well presented and interpreted. Also, the experimental manufacturing of the finished product is presented and correlations with the simulations are done.
The conclusions present very well the obtained results and observed correlations by considering tool design, tool manufacturing, injection molding process, and product quality.
I have only one suggestion:
Page 8, Figure 4: In the notations from “a” to “f “, the “d” notation is missing, but is mentioned at line 316.
Best regards,
A reviewer
Author Response
Dear reviewer
Thank you for your kind review and interest in our work.
Figure 4 and the corresponding text have been corrected so that the notation is now consistent.
Yours sincerely
Reviewer 2 Report
The authors reported “A holistic approach to cooling system selection and injection molding process optimization based on non-dominated sorting” . The manuscript comes the following comments.
1- The abstract must report the optimized conditions and efficacy obtained together with the significance of the model developed .
2- Line 28-43; this statement is too long without citation by which those information were obtained from ; pls cite
3- Pay attention to intext referencing ; Line 71-73; 80-89 Tubes with Ø 2 to Ø 8 mm and inserts with Ø 2 to Ø 10 mm are used.
4- Figure 2 and 3 ; systematic representation seems confusion , pls show some arrows
5- the process needs to be represented
6- Table 2 check the consistency of the decimal places
Check consistency of the references
Round 2
Reviewer 2 Report
Thanks for incorporated the comments